# Comparison of Automated Keratometer and Scheimpflug Tomography for Predicting Refractive Astigmatism in Pseudophakic Eyes

**DOI:** 10.3390/diagnostics13243687

**Published:** 2023-12-18

**Authors:** Kyung-Sun Na, Giacomo Savini, Woong-Joo Whang, Kristian Næser

**Affiliations:** 1Department of Ophthalmology, Yeouido St. Mary’s Hospital, College of Medicine, The Catholic University of Korea, Seoul 16247, Republic of Korea; githen@hanmail.net; 2IRCCS—G.B. Bietti Foundation, 00184 Rome, Italy; dr.giacomo.savini@gmail.com; 3Randers Regional Hospital, 8930 Randers, Denmark; krisnaes@rm.dk

**Keywords:** astigmatism, corneal curvature, Scheimpflug tomography

## Abstract

Purpose: To analyse the correspondence between refractive astigmatism and corneal astigmatism in pseudophakic eyes with non-toric intraocular lenses. Setting: Yeouido St. Mary hospital, Seoul, Republic of Korea. Design: Evaluation of a diagnostic test instrument. Methods: This retrospective study included 95 eyes of 95 patients. Corneal astigmatism was measured with an automated keratometer (RK-5, Canon) and Scheimpflug tomography (Pentacam HR, Oculus). Refractive astigmatism was compared to keratometric astigmatism (based on anterior corneal measurements only), equivalent K-reading, and total corneal astigmatism (both based on anterior and posterior corneal measurements). Vector analysis was carried out by Næser’s polar value method. The accuracy was defined as the average magnitude of the vectorial difference in astigmatism (DA). Each corneal measurement was optimized in retrospect by a multiple linear regression equation between refractive and corneal astigmatism. Results: Keratometric astigmatism overestimated with-the-rule (WTR) refractive astigmatism and underestimated against-the-rule (ATR) refractive astigmatism. Several measurements based on both corneal surfaces’ values did not show any statistically significant difference with respect to refractive astigmatism. The mean corneal astigmatism by total corneal refractive power (TCRP) at 4.0 mm (zone/pupil) produced the lowest mean arithmetic DA and the highest percentage of eyes with a DA ≤ 0.50 dioptre. After optimization, the accuracies of automated KA and TCRP 4.0 mm (zone/pupil) were similar. Conclusions: Total corneal astigmatism measured by Scheimpflug tomography at a 4.0 mm zone centered on the pupil accurately reflects the refractive astigmatism in pseudophakic eyes. However, the accuracy of total corneal astigmatism is not different from automated KA after optimization.

## 1. Introduction

Corneal power and astigmatism have traditionally been determined using instruments that measure only the anterior corneal curvature. The total corneal power and astigmatism are subsequently estimated by means of the keratometric index (usually 1.3375), which takes into account the negative power of the posterior corneal surface. Therefore, the keratometric index can be used to assess the toral corneal power. More than 30% of eyes display > 1.0 dioptre (D) of keratometric astigmatism (KA), which can limit the patient’s postoperative uncorrected visual acuity if not treated at the time of cataract surgery [1,2]. Thus, selecting the best method to measure corneal astigmatism (CA) is very important for minimising postoperative residual refractive astigmatism (RA).

Corneal astigmatism is a common refractive error caused by an irregular shape of the cornea. This results in blurred or distorted vision. Miscalculation of the power of a toric intraocular lens (IOL) can lead to several postoperative complications and side effects. Residual astigmatism induces reduced visual quality, patient dissatisfaction, and the need for additional procedures.

The introduction of Scheimpflug tomography made it possible to measure the posterior corneal curvature and directly obtain the total corneal power and astigmatism (without relying on fictitious assumptions related to the keratometric index) [3]. In normal eyes, posterior corneal astigmatism (PCA) mean values have been reported to range from 0.30 to 0.54 D, but the magnitude can be higher than 1 D in 5.7% of eyes [4,5]. As KA already accounts for the PCA through its keratometric index, the mean difference between total corneal astigmatism (TCA) and KA is lower (0.20–0.25 D) than the mean magnitude of PCA [6,7]. In good agreement with these data, corneal astigmatism as determined by TCA measurements corresponded better with manifest refractive cylinder than corneal astigmatism based on KA in phakic eyes [8].

To determine the Scheimpflug tomography measurement that most accurately predicts the RA, we designed a study on non-toric pseudophakic eyes, where the influence of lenticular astigmatism can be excluded. Under the assumption that the TCA must be equal to the RA in these eyes, we compared manifest RA and different corneal astigmatism measurements, with the goal of determining which value is most closely associated with the refractive cylinder and thus may be considered as the best option to calculate the power of toric intraocular lenses (IOLs).

## 2. Materials and Methods

### 2.1. Patients and Surgical Procedures

This retrospective study included a consecutive series of pseudophakic patients enrolled between November 2015 and January 2021. Each patient signed an informed consent document prior to enrolment, and all study methods adhered to the tenets described in the Declaration of Helsinki. The study protocol was reviewed and approved by the Institutional Review Board of Yeouido St. Mary Hospital (Seoul, Republic of Korea). Exclusion criteria were a history of any ocular disease, previous ocular surgery (other than cataract surgery), and occurrence of intraoperative and/or postoperative complications; corrected distance visual acuity had to be higher than 20/40. Eyes with any degree of corneal astigmatism were enrolled in the study.

Phacoemulsification was performed through a 2.2-mm temporal incision, and a non-toric intraocular lens (IOL, ZCB00, Johnson & Johnson Vision, Irvine, CA, USA) was implanted into the bag.

### 2.2. Postoperative Measurements

Measurements of RA and CA were performed six months after the surgery. Manifest refraction was measured in a plus-cylinder format using an automated refractometer with a vertex distance setting of zero mm (Canon RK-5, Tokyo, Japan), which was subsequently refined by subjective refraction. CA was assessed by measuring each eye once with an automated keratometer (Canon RK-5, Tokyo, Japan) and Scheimpflug tomography (Pentacam HR, software version 1.17r91; Oculus, Wetzlar, Germany). This was used to analyse the cornea via a 25-picture scan; only scans whose quality specification was graded as “OK” by the instrument software were included.

In addition to keratometric astigmatism (KA) using an automated keratometer, the following corneal power measurements using Scheimpflug tomography were evaluated:

*Keratometric astigmatism (KA)*: This value represents the difference between the power of the steep and flat meridians. It is equivalent to the simulated K-value used in traditional corneal topography and is calculated by entering the corneal curvature radius into a thin-lens formula for paraxial imagery, which considers the cornea as a single refractive sphere. The corneal radii were converted into dioptric power values using the keratometric index of refraction (1.3375).

*Total corneal refractive power (TCRP)*: This value corresponds to TCA, and it is automatically measured by ray tracing and calculated using the values for anterior corneal radius, posterior corneal radius, and corneal thickness. Snell’s law and the specific refractive indices of air, cornea, and aqueous humour were used to calculate the corneal power. Measurements at a 2.0-mm zone/ring, 3.0-mm zone/ring, and 4.0-mm zone/ring centred on both the pupil and the apex were considered for the statistical analysis.

IOL decentration and tilt were evaluated using Scheimpflug tomography, and cases in which IOL decentration was >0.4 mm or tilt was >5° were excluded from the study. IOL decentration and tilt were measured according to methods previously described by de Castro et al. [9] and Kranitz et al. [10]. IOL decentration was determined as the distance between the IOL centre and the pupillary axis. Positive and negative horizontal decentration indicate temporal and nasal decentration, respectively, while positive and negative vertical decentration indicate superior and inferior decentration, respectively. In this study, we calculated the mean absolute value of IOL decentration. Total decentration is the magnitude of the vector resulting from the horizontal and vertical decentration. A positive IOL tilt around the x-axis indicates that the superior edge of the IOL is located on the front, while a positive tilt around the y-axis indicates that the temporal edge of the IOL moves backward when compared with the nasal edge. We also calculated the mean absolute value of IOL tilt.

### 2.3. Astigmatism Analysis

Vectorial astigmatism analysis was performed using the Næser polar value method [11,12]. Specific calculations were performed as previously reported by Bregnhøj et al. [13].

CA and RA can be described as two-dimensional vectors. For this purpose, the net astigmatism (M @ α), where M (M ≥ 0) is the astigmatic magnitude in dioptres (D) and α is the steep astigmatic direction in degrees (°), was transformed into two polar values in units of dioptres in the following general format [12]:Polar value along Φ = KP(Φ) = M × cos(2 × (α − Φ))(1)
Polar value along (Φ + 45) = KP(Φ + 45) = M × sin(2 × (α − Φ))(2)
where Φ is the reference meridian (°). In the present study, we used a *variable* reference meridian Φ = steeper CA α to describe RA as a function of CA [13]. Therefore, the reference meridian Φ varied for each corneal measurement modality and for each corneal measurement.

KP(Φ) is positive for astigmatic meridians α along plane Φ and negative for meridians along Φ + 90. KP(Φ + 45) is positive for astigmatic meridians α rotated in a counter-clockwise direction and negative for a clockwise rotation relative to Φ.

The difference in astigmatism (DA), that is, the difference between refractive and corneal astigmatism, was separately calculated for both polar values: ∆KP(Φ) = KP(Φ)_RA_ − KP(Φ)_CA_(3)
 ∆KP(Φ + 45) = KP(Φ + 45)_RA_ − KP(Φ + 45)_CA_(4)

ΔKP(Φ) is positive for CA underestimation of RA and negative for overestimation. ΔKP(Φ + 45) is positive for RA meridian rotated in a counter-clockwise direction and negative for a clockwise rotation relative to Φ.

The DA was reconverted to the standard net cylinder format using the following equations:(5)$$Magnitude\;of\;DA=\sqrt{{\Delta KP(\Phi )}^{2}+{\Delta KP(\Phi +45)}^{2}}$$
(6)$$Angle=arctan\;(\frac{M-∆KP(\Phi )}{\Delta KP(\Phi +45)})$$

In this context, Equation (6) provides the meridian relative to the steep anterior corneal meridian. The average net astigmatism was calculated using the average values for KP(Φ) and KP(Φ + 45). The average DA magnitude (not considering the meridian) was calculated by inserting the individual values in Equation (5), followed by an averaging procedure.

A low mean value of DA indicates a high accuracy, with the corneal astigmatism measurement well aligned in both meridian and magnitude, and the corneal plane manifesting as a refractive cylinder. Aggregate analysis of astigmatism was also performed using the astigmatism double angle polar tool [14].

All eyes were divided into three groups according to the orientation of the steep meridian (as measured with an automated keratometer), and the data for each group were separately analysed. In the with-the-rule (WTR) astigmatism group, the orientation of the steep meridian ranged from 61° to 120°, whereas in the against-the-rule (ATR) astigmatism group, it ranged from 151° to 180° or from zero to 30°. The cases with astigmatism meridian not belonging to the above two groups were classified as the oblique astigmatism group (31° to 60° or 121° to 150°).

We calculated polar values along the variable meridians to investigate each corneal measurement modality for all astigmatic directions. This approach allowed us to differentiate between the most accurate (with the smallest DA values) corneal measurements.

Finally, we performed optimization of the corneal measurements based on multiple linear regression analysis. As previously reported, this optimization followed the general format [13]:KP(Φ)_RA_ = a + b × KP(Φ)_KA_ + c × cos(2α)(7)

### 2.4. Statistical Analysis

All statistical analyses were performed using IBM SPSS Statistics for Windows (version 21.0. IBM Corp., Armonk, NY, USA). The normality of the data distribution was assessed using the Kolmogorov–Smirnov test. As the data were not universally normally distributed, non-parametric methods were consistently employed. One- and two-sample Wilcoxon signed-rank tests were performed to determine differences between refractive cylinder and corneal astigmatism and to evaluate whether the mean values of ∆KP(Φ) and ∆KP(Φ + 45) were significantly different from zero. The Friedman test was also performed to determine differences in cylinder magnitude and polar values among all astigmatism measurements. Multiple linear regression tests were performed to build equations predicting KP(Φ)_RA_. Statistical significance was set at *p* < 0.05.

## 3. Results

A total of 95 left eyes (95 patients, 41 males and 54 females; mean age 64.09 ± 9.20 years; range, 39–83 years) were evaluated. Corneal astigmatism was classified as WTR in 45 eyes (47.4%), ATR in 34 eyes (35.8%), and oblique in 16 eyes (16.8%). All 95 eyes revealed IOL decentration less than 0.4 mm and IOL tilt less than 4° (Table 1).

### 3.1. Comparing Refractive and Corneal Magnitudes of Astigmatism

The average magnitudes (calculated without vector analysis) for corneal and refractive astigmatism were generally similar across the corneal measurement modalities (Table 2).

### 3.2. Comparing Refractive and Corneal Astigmatism Using the Steep Corneal Meridian as the Reference Plane

Table 3 reports the DA between RA and all CA measurement modalities for the whole sample. According to Friedman’s test, the ∆KP(Φ) values revealed a statistically significant difference (*p* < 0.001). The sign for ΔKP(Φ) was negative for all corneal measurements. Therefore, the corneal measurements averagely overestimated the RA, and all values of ∆KP(Φ) were significantly different from zero (*p* < 0.01). In the total group, generally, the average ΔKP(Φ + 45) was clinically non-significant and close to zero (Table 3). This implies that the magnitudes of the refractive and the different corneal astigmatisms were different, while their directions were similar. Consequently, the orientation of the net cylinders was approximately at 90 degrees for all corneal measurements. The mean DA magnitude, the average of the individually calculated absolute differences in RA and CA astigmatism without considering the meridian, amounted to 0.51 ± 0.25 D for automated keratometry. The lowest mean DA for corneal measurements by the Scheimpflug camera (0.42 ± 0.24 D) was achieved by TCRP pupil/zone at 4.0 mm.

For the WTR subgroup, the average ΔKP(Φ) was negative and significantly different from zero for all corneal measurement modalities (Table 4). Automated keratometry and TCRP pupil/zone at 4 mm overestimated refraction with mean values of 0.44 D and 0.26 D, respectively. The average ΔKP(Φ) was close to zero for all corneal measuring modalities in the small group of eyes with oblique astigmatism (Table 5).

For eyes with ATR astigmatism, mean ΔKP(Φ) amounted to 0.26 D for KA and −0.05 D for TCRP pupil/zone at 4.0 mm (Table 6).

Figure 1 shows the magnitudes of DA in the whole sample and the three subgroups. The measurements achieving the most accurate outcomes across all three subgroups were the TCRP pupil/zone values at 4.0 mm, which consistently produced mean magnitudes of DA ≤ 0.5 D. Since the value at 4.0 mm also provided the highest (or second highest) percentage of cases with a DA magnitude ≤ 0.5 D, ranging from 55.6% to 68.8%, it can be considered the most accurate corneal measurement of astigmatism. Automated KA was more accurate than Scheimpflug KA in the entire group and in all subgroups.

The optimization formulae for all corneal measurements are shown in Table 7. The regression constants related to cos(2α) were larger for the two KA measurements compared to the remaining corneal measurements, which all included the posterior CA. However, as none of these constants was zero, they required some correction for direction-based errors. The use of these optimized formulae led to an average ΔKP(Φ) error of zero and reduced standard deviations for all corneal measurements (Table 8). The ΔKP(Φ+45) was not optimized, and was therefore identical to the values reported in Table 3. The optimized TCRP pupil/zone at 4.0 mm value was still the most accurate measurement, with a mean DA magnitude of 0.38 (±0.21) D and 73.7% of DA magnitudes within 0.5 D. The average 0.38 (±0.23) D accuracy for optimized automated keratometry was of similar magnitude and was not statistically significantly different (Table 9). The ΔKP(Φ) values of both parameters showed statistical differences before and after optimization. The mean DA magnitude was significantly reduced after the optimization of the automated KA (Table 9). The accuracies as a result of the optimization therefore improved significantly for automated KA, but only moderately for TCRP pupil/zone at 4.0 mm. The bivariate plot with 95% confidence ellipses for automated KA and TCRP at 4.0 mm apex/zone is shown in Figure 2 and Figure 3.

## 4. Discussion

In the present study, we found that several TCA values were in good agreement with RA. Moreover, we confirmed that corneal astigmatism values based on the curvature of both corneal surfaces (i.e., TCA) resembled RA more closely than those based on the anterior corneal curvature only (i.e., KA).

Ideally, corneal astigmatism measurement should provide us with consistently good results in eyes with WTR, ATR, and oblique astigmatism. In this regard, the value that gave the best result was the TCRP at 4.0 mm (zone/pupil). However, the residual astigmatism is small but not negligible, and should be considered when planning toric IOL implantation. A possible explanation may be related to imperfect measurements of corneal astigmatism by Scheimpflug tomography or to other unknown sources of error.

We found that the arithmetic mean DA determined by automated KA was lower than that determined by KA based on Scheimpflug tomography (Table 3, Table 4, Table 5 and Table 6). This result is similar to that of a previous study [15]. The relatively lower accuracy of Scheimpflug tomography may be because of the prolonged time required for image acquisition [16], due to the different diameters of the analysed zone or the different technologies used to measure the corneal curvature.

In the analysis on the whole cohort using the variable steep anterior corneal meridian as reference, the Scheimpflug TCRP measurement at 4.0 mm pupil/zone was the most accurate. Therefore, the present study suggests that this modality should be used in all Pentacam measurements to predict refractive astigmatism. The accuracy of this measurement improved only moderately as a result of optimization (Table 9), further strengthening the case for a physiological representation of refractive astigmatism. In contrast, the accuracy of automated KA improved significantly during optimization, thereby reaching a similar accuracy as the optimized TCRP at 4.0 mm pupil/zone. For this group of patients and with these specific measuring devices, the accuracies of optimized automated KA and TCRP at 4.0 mm (pupil/zone) are therefore identical. This may not be true for other keratometers.

Our findings relative to TCA and KA are in good agreement with previous studies showing that TCA measurements mirror RA better than KA. For example, the accuracy of TCA (3.0 mm), as measured with both a Scheimpflug rotating camera and a colour light-emitting diode corneal topographer, was higher than that of KA in pseudophakic eyes with non-toric IOLs [17]. Similarly, the prediction error in RA was lower with TCA (3.0 mm) than with KA in a sample of eyes implanted with toric IOLs [18]. It will be interesting to see if TCA at 4.0 mm (which was not analysed in previous studies) will further improve the outcomes of toric IOL implantation.

In the present study, the measured KA values were optimized using a multiple linear regression formula, thereby predicting KP(Φ)_RA_ using KP(Φ)_KA_ and cos(2α), where α is the meridian of the steep corneal meridian (Table 7).
KP(Φ)_RA_ = −0.04 + 0.92 × KP(Φ)_KA_+ 0.39 × cos(2α)    KP(Φ)_RA_ = −0.04 + 0.94 × KP(Φ)_KA_+ 0.39 × cos(2α)(8)(when corneal refractive index = 1.3315).

This model assumes that the directions of RA and CA are identical, as evidenced by the average values for ΔKP(Φ+45)_RA_ close to zero (as shown in Table 3, the average ΔKP(Φ+45)_RA_ amounted to only −0.04 D). This correlation allows for an individual estimation of KP(Φ)_RA_ for any combination of KA magnitude and meridian, that is, for both eyes with WTR, oblique, and ATR astigmatism. For the WTR KA of 1.0 D along 90°, KP(Φ)_RA_ is predicted as 0.49 D and, therefore, the net RA net astigmatism as 0.49 D along 90°. The RA net astigmatism is predicted as 1.27 D along zero degrees for the ATR net cylinder 1.0 D along zero degrees. This correlation was developed in a previous study, based on a pseudophakic eye model of 184 eyes with a postoperative KA of 0.91 D, using a keratometric index of 1.3375 [13] as KP(Φ)_RA_ = −0.09 + 0.68 × KP(Φ)_KA_ + 0.33 × cos(2α). In that study, the regression constant for KP(Φ)_KA_ was considerably lower than the similar value in this study, 0.68 versus 0.92, which might be explained by the lower KA values in the present study (Table 2). In the present study, the regression constants for cos(2α) were quite similar, 0.33 versus 0.39, in the present study. The same methodology was used to optimize KA for toric IOL calculation [18], yielding the regression equation 0.103 + 0.836 × KA + 0.457 × cos2α. This equation is quite similar to the present correlation for KA, although the toric IOL optimization is far more complex with additional variables, such as exact vergence calculation and toric IOL effective power. Measurement precision is of the most importance in all optimization procedures, and might be improved by using the average of multiple, rather than single, determinations of refractive and corneal astigmatism. The necessity for regression constants of the optimized Scheimpflug tomography measurements in Table 7 is a sign of imperfections of the measured values. These may be due to internal calculation algorithms, erroneous measurement methods, or unknown factors.

The present study has several strengths and limitations. Among the former, we ruled out confounding factors induced by lenticular astigmatism as well as by tilt and decentration; therefore, only pseudophakic eyes with a decentration <0.4 mm and a tilt <5 degrees were enrolled (the critical amounts of decentration and tilt have been reported to be 0.4~0.8 mm and 5~10 degrees, respectively) [19,20]. Furthermore, we used Scheimpflug tomography, whose reproducibility when taking measurements of IOL decentration and tilt has been shown to be high [9,21]. This is the first study to evaluate pseudophakic eyes with the effect of lenticular astigmatism minimized, which is different from previous studies that have studied phakic eyes without cataracts [8]. Regarding limitations, we did not measure TCA with other devices or technologies, which may offer interesting comparisons.

In conclusion, TCA measurements using Scheimpflug tomography were closer to the mean RA with respect to corneal astigmatism measurements derived from the anterior corneal surface only (i.e., KA). Among the several available options, the TCRP value centred on the pupil and including a 4.0-mm diameter zone was the one that most closely mirrored RA in pseudophakic eyes. Automated KA measurements provided similar accuracies after optimization. Thus, both values may be used for accurate toric IOL calculation. The results of this study are expected to aid in corneal assessment prior to astigmatism-correcting cataract surgery and in surgical planning, including identifying indications for toric IOLs and determining cylindric power.

## Figures and Tables

**Figure 1 diagnostics-13-03687-f001:**
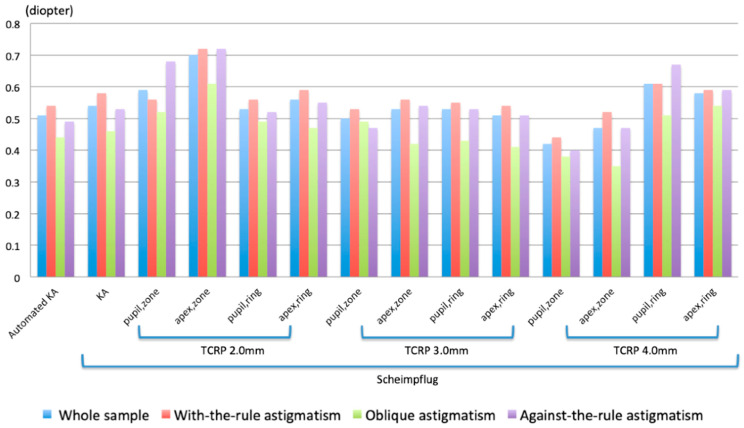
Mean DA magnitudes in a whole sample and three subgroups divided according to the orientation of the steep meridian.

**Figure 2 diagnostics-13-03687-f002:**
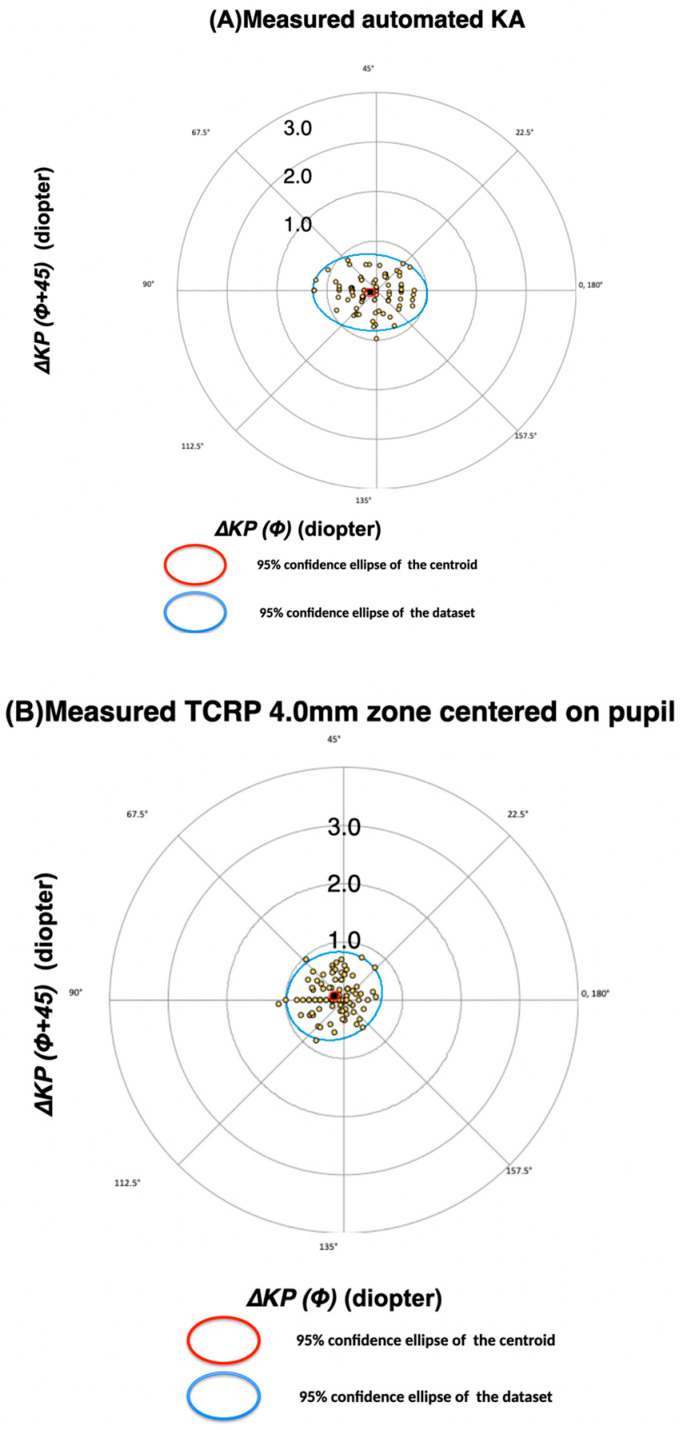
Difference between refractive and corneal astigmatism on double angle polar plots *before* optimization. Individual values and combined means (centroids) with their 95% confidence ellipses. DAs were derived from (**A**) the automated keratometer and (**B**) Scheimpflug TCRP 4.0 mm zone centred on pupil.

**Figure 3 diagnostics-13-03687-f003:**
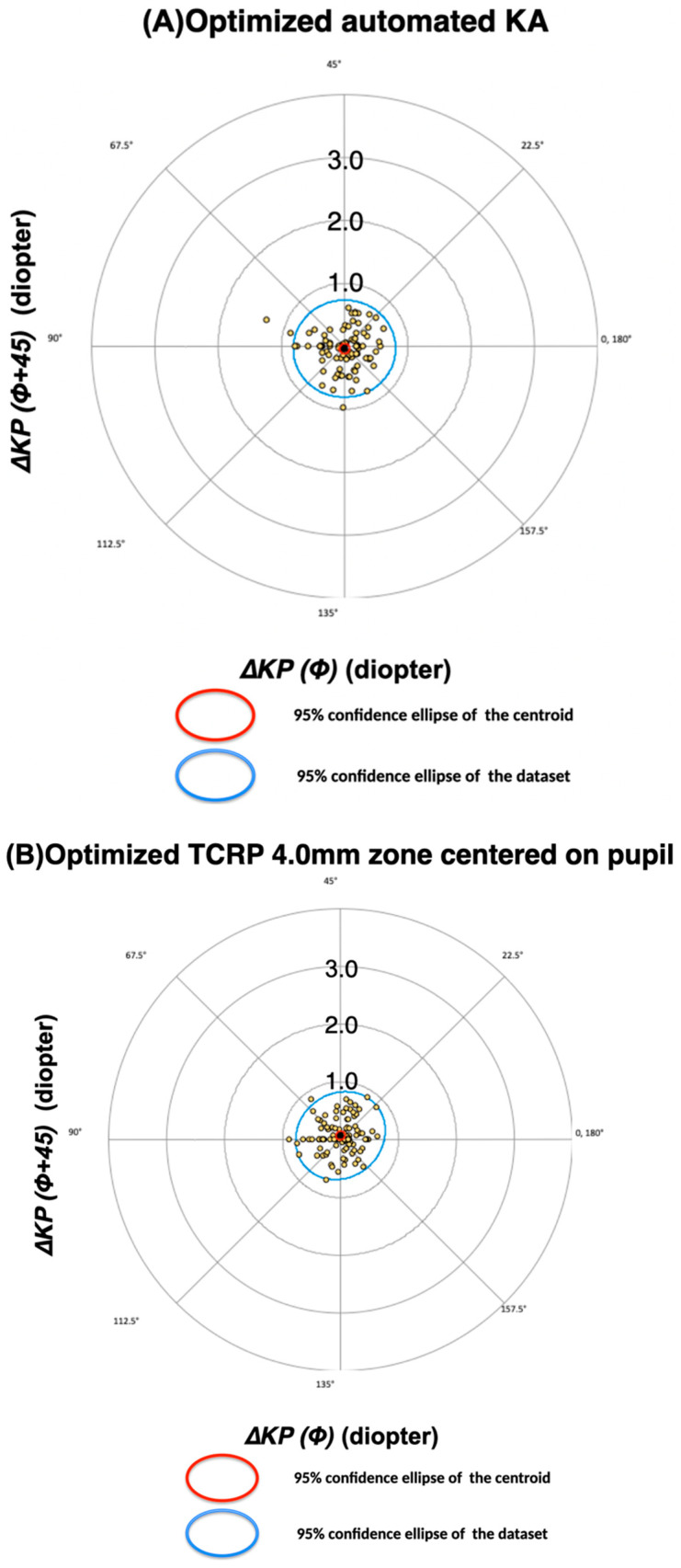
Difference between refractive and corneal astigmatism on double angle polar plots *after* optimization. Legend as for Figure 1. Following optimization, the cluster of points and centroids were more centred on the coordinate origin, and the extension of the confidence ellipses diminished, thereby signifying increased accuracy.

**Table 1 diagnostics-13-03687-t001:** Intraocular lens positioning in 95 eyes.

	Absolute Value	
	Mean± Standard Deviation	Range
Horizontal decentration (mm)	0.16 ± 0.11	0~0.38
Vertical decentration (mm)	0.14 ± 0.11	0~0.35
Total decentration (mm)	0.24 ± 0.12	0~0.39
Horizontal tilt (degrees)	1.25 ± 0.95	0~3.0
Vertical tilt (degrees)	1.46 ± 1.08	0~3.5

**Table 2 diagnostics-13-03687-t002:** Mean values for refractive astigmatism and corneal astigmatism in the whole sample (*n* = 95 eyes). Calculations are based on magnitudes of astigmatism without considering direction of the astigmatism.

Parameter		Cylinder Magnitude (D) ± SD	Difference with RA	*p* Value *
RA		0.70 ± 0.61		
Automated K		0.72 ± 0.56	−0.02 ± 0.45	0.845
Scheimpflug KA		0.73 ± 0.56	−0.03 ± 0.42	0.576
Scheimpflug TCRP 2.0 mm	pupil/zone	0.78 ± 0.60	−0.08 ± 0.47	0.148
	apex/zone	1.03 ± 0.67	−0.33 ± 0.50	<0.001
	pupil/ring	0.87 ± 0.64	−0.17 ± 0.41	0.001
	apex/ring	0.86 ± 0.61	−0.16 ± 0.41	0.001
Scheimpflug TCRP 3.0 mm	pupil/zone	0.84 ± 0.61	−0.14 ± 0.39	0.233
	apex/zone	0.80 ± 0.59	−0.10 ± 0.40	0.033
	pupil/ring	0.80 ± 0.55	−0.10 ± 0.39	0.023
	apex/ring	0.83 ± 0.64	−0.13 ± 0.41	0.007
Scheimpflug TCRP 4.0 mm	pupil/zone	0.75 ± 0.60	−0.05 ± 0.35	0.300
	apex/zone	0.75 ± 0.57	−0.04 ± 0.37	0.357
	pupil/ring	0.81 ± 0.55	−0.11 ± 0.45	0.037
	apex/ring	0.81 ± 0.59	−0.11 ± 0.45	0.064
Difference between parameters (*p* value) **		<0.001		

SD: standard deviation; RA: refractive astigmatism; KA: anterior K; TCRP: total corneal refractive power. * *p* value by the Wilcoxon signed ranked test comparing average value for each corneal measurement with refractive astigmatism, ** *p* value by the Friedman test comparing the average values of all corneal measurements.

**Table 3 diagnostics-13-03687-t003:** Difference in astigmatism (DA) derived from the refractive cylinder and *measured value* of corneal astigmatism in the whole sample (*n* = 95). Calculations are based on polar values along *variable* meridians in the form of the steep anterior corneal meridian for each eye and for each measuring modality.

Keratometric Measurement		∆KP(Φ) (D) ± SD	Difference from Zero (*p* Value) †	∆KP(Φ+45) (D) ± SD	Difference from Zero (p Value) †	Mean DA (D @ Degree)	Mean DA Magnitude (D) ± SD	Range of DA (D)	Percentage of DA Magnitude within 0.50 D
Automated KA		−0.13 ± 0.46	0.005	−0.04 ± 0.31	0.579	0.14 @ 98	0.51 ± 0.25	0–1.25	51.6
Scheimpflug KA		−0.22 ± 0.40	<0.001	−0.01 ± 0.37	0.799	0.22 @ 91	0.54 ± 0.23	0–1.20	46.3
Scheimpflug TCRP 2.0 mm	pupil/zone	−0.28 ± 0.49	<0.001	0.04 ± 0.42	0.189	0.28 @ 86	0.59 ± 0.37	0.09–2.03	49.5
	apex/zone	−0.52 ± 0.51	<0.001	0.06 ± 0.37	0.077	0.52 @ 87	0.70 ± 0.42	0.07–2.03	37.9
	pupil/ring	−0.33 ± 0.40	<0.001	0.08 ± 0.32	0.017	0.34 @ 84	0.53 ± 0.31	0–1.50	52.6
	apex/ring	−0.34 ± 0.39	<0.001	0.10 ± 0.37	0.007	0.35 @ 82	0.56 ± 0.31	0.02–1.50	49.5
Scheimpflug TCRP 3.0 mm	pupil/zone	−0.29 ± 0.38	<0.001	0.07 ± 0.32	0.018	0.30 @ 83	0.50 ± 0.29	0.05–1.30	54.7
	apex/zone	−0.27 ± 0.38	<0.001	0.11 ± 0.36	0.004	0.29 @ 79	0.53 ± 0.29	0.05–1.30	55.8
	pupil/ring	−0.26 ± 0.39	<0.001	0.13 ± 0.34	0.001	0.29 @ 77	0.53 ± 0.28	0.03–1.38	51.6
	apex/ring	−0.27 ± 0.40	<0.001	0.11 ± 0.32	0.002	0.29 @ 79	0.51 ± 0.29	0–1.39	54.7
Scheimpflug TCRP 4.0 mm	pupil/zone	−0.16 ± 0.33	<0.001	0.07 ± 0.30	0.013	0.18 @ 78	0.42 ± 0.24	0.03–1.10	62.1
	apex/zone	−0.19 ± 0.37	<0.001	0.10 ± 0.33	0.007	0.21 @ 76	0.47 ± 0.26	0.02–1.07	51.6
	pupil/ring	−0.34 ± 0.47	<0.001	0.08 ± 0.38	0.059	0.35 @ 83	0.61 ± 0.34	0.03–1.40	43.2
	apex/ring	−0.31 ± 0.45	<0.001	0.05 ± 0.39	0.338	0.31 @ 85	0.58 ± 0.33	0.07–1.44	45.3
Difference between parameters (*p* value) *		<0.001		0.027			<0.001		

SD: standard deviation; KA: keratometric astigmatism; TCRP: total corneal refractive power; * *p* value by the Friedman test comparing average values of all corneal measurements; † *p* value by one-sample Wilcoxon signed rank test.

**Table 4 diagnostics-13-03687-t004:** Difference in astigmatism (DA) derived from the refractive cylinder and *measured value* of corneal astigmatism in the with-the-rule astigmatism (*n* = 45). Calculations are based on polar values along *variable* meridians in the form of the steep anterior corneal meridian for each eye and for each measuring modality.

Keratometric Measurement		∆KP(Φ) (D) ± SD	Difference from Zero (*p* Value) †	∆KP(Φ+45) (D) ± SD	Difference from Zero (*p* Value) †	Mean DA (D @ Degree)	Mean DAMagnitude (D) ± SD	Range of DA (D)	Percentage of DA within 0.50 D
Automated KA		−0.44 ± 0.32	<0.001	−0.02 ± 0.28	0.905	0.44 @ 0	0.54 ± 0.28	0.08–1.25	46.7
Scheimpflug KA		−0.47 ± 0.28	<0.001	0.01 ± 0.33	0.915	0.38 @ 0	0.58 ± 0.26	0.09–1.20	40.0
Scheimpflug TCRP 2.0 mm	pupil/zone	−0.42 ± 0.37	<0.001	0.11 ± 0.31	0.029	0.06 @ 52	0.56 ± 0.32	0.09–1.50	55.6
	apex/zone	−0.61 ± 0.42	<0.001	0.12 ± 0.33	0.041	0.01 @ 169	0.72 ± 0.38	0.16–1.95	33.3
	pupil/ring	−0.43 ± 0.35	<0.001	0.11 ± 0.29	0.023	0.07 @ 34	0.56 ± 0.30	0.05–1.40	51.1
	apex/ring	−0.47 ± 0.33	<0.001	0.15 ± 0.31	0.002	0.05 @ 49	0.59 ± 0.30	0.07–1.44	44.4
Scheimpflug TCRP 3.0 mm	pupil/zone	−0.40 ± 0.33	<0.001	0.11 ± 0.29	0.021	0.07 @ 28	0.53 ± 0.28	0.15~1.30	55.6
	apex/zone	−0.43 ± 0.31	<0.001	0.15 ± 0.32	0.005	0.06 @ 35	0.56 ± 0.27	0.09–1.30	51.1
	pupil/ring	−0.38 ± 0.40	<0.001	0.10 ± 0.28	0.019	0.08 @ 11	0.55 ± 0.29	0.05–1.18	46.7
	apex/ring	−0.36 ± 0.41	<0.001	0.09 ± 0.28	0.027	0.12 @ 7	0.54 ± 0.30	0.09–1.31	51.1
Scheimpflug TCRP 4.0 mm	pupil/zone	−0.26 ± 0.34	<0.001	0.08 ± 0.28	0.020	0.11 @ 11	0.44 ± 0.26	0.05–1.10	55.6
	apex/zone	−0.33 ± 0.36	<0.001	0.10 ± 0.30	0.043	0.10 @ 15	0.52 ± 0.27	0.09–1.03	48.9
	pupil/ring	−0.45 ± 0.44	<0.001	0.06 ± 0.30	0.240	0.19 @ 1	0.61 ± 0.34	0.10–1.40	40.0
	apex/ring	−0.41 ± 0.45	<0.001	0.04 ± 0.30	0.464	0.23 @ 179	0.59 ± 0.33	0.07–1.40	44.4
Difference between parameters (*p* value) *		<0.001		0.030			<0.001		

SD: standard deviation; KA: keratometric astigmatism; TCRP: total corneal refractive power; * *p* value by the Friedman test comparing average values of all corneal measurements; † *p* value by one-sample Wilcoxon signed rank test.

**Table 5 diagnostics-13-03687-t005:** Difference in astigmatism (DA) derived from the refractive cylinder and *measured value* of corneal astigmatism in the oblique astigmatism (*n* = 16). Calculations are based on polar values along *variable* meridians in the form of the steep anterior corneal meridian for each eye and for each measuring modality.

Keratometric Measurement		∆KP(Φ) (D) ± SD	Difference from Zero (*p* Value) †	∆KP(Φ+45) (D) ± SD	Difference from Zero (*p* Value) †	Mean DA (D @ Degree)	Mean DA Magnitude (D) ± SD	Range of DA (D)	Percentage of DA within 0.50 D
Automated KA		−0.10 ± 0.29	0.148	−0.09 ± 0.38	0.469	0.29 @ 172	0.44 ± 0.20	0.05–0.97	68.8
Scheimpflug KA		−0.19 ± 0.31	0.028	0.03 ± 0.38	0.753	0.31 @ 169	0.46 ± 0.23	0.00–0.84	62.5
Scheimpflug TCRP 2.0 mm	pupil/zone	−0.09 ± 0.51	0.642	−0.04 ± 0.39	0.433	0.16 @ 4	0.52 ± 0.38	0.12–1.25	50.0
	apex/zone	−0.38 ± 0.58	0.039	0.04 ± 0.32	0.875	0.08 @ 157	0.61 ± 0.44	0.07–1.53	50.0
	pupil/ring	−0.27 ± 0.46	0.061	0.06 ± 0.27	0.666	0.05 @ 136	0.49 ± 0.34	0.00–1.19	56.3
	apex/ring	−0.21 ± 0.42	0.063	0.09 ± 0.33	0.470	0.09 @ 167	0.47 ± 0.32	0.02–1.11	56.3
Scheimpflug TCRP 3.0 mm	pupil/zone	−0.28 ± 0.44	0.049	0.08 ± 0.27	0.382	0.02 @ 159	0.49 ± 0.33	0.06~1.16	50.0
	apex/zone	−0.12 ± 0.41	0.278	0.07 ± 0.29	0.510	0.10 @ 6	0.42 ± 0.30	0.05~1.01	62.5
	pupil/ring	−0.20 ± 0.32	0.026	0.12 ± 0.29	0.158	0.02 @ 151	0.43 ± 0.21	0.14–0.88	62.5
	apex/ring	−0.18 ± 0.29	0.023	0.09 ± 0.29	0.328	0.04 @ 133	0.41 ± 0.19	0.15–0.73	68.8
Scheimpflug TCRP 4.0 mm	pupil/zone	−0.14 ± 0.29	0.079	0.10 ± 0.30	0.365	0.08 @ 169	0.38 ± 0.22	0.07–0.80	68.8
	apex/zone	−0.10 ± 0.33	0.098	0.10 ± 0.26	0.198	0.07 @ 5	0.35 ± 0.25	0.03–0.88	62.5
	pupil/ring	−0.37 ± 0.28	0.001	0.13 ± 0.39	0.638	0.11 @ 4	0.51 ± 0.35	0.03–1.30	62.5
	apex/ring	−0.29 ± 0.30	0.003	0.01 ± 0.49	0.730	0.28 @ 169	0.54 ± 0.34	0.10–1.44	50.0
Difference between parameters (*p* value) *		0.026		0.389			0.330		

SD: standard deviation; KA: keratometric astigmatism; TCRP: total corneal refractive power; * *p* value by the Friedman test comparing average values of all corneal measurements; † *p* value by one-sample Wilcoxon signed rank test.

**Table 6 diagnostics-13-03687-t006:** Difference in astigmatism (DA) derived from the refractive cylinder and *measured value* of corneal astigmatism in the against-the-rule astigmatism (*n* = 34). Calculations are based on polar values along *variable* meridians in the form of the steep anterior corneal meridian for each eye and for each measuring modality.

Keratometric Measurement		∆KP(Φ) (D) ± SD	Difference from Zero (*p* Value) †	∆KP(Φ+45) (D) ± SD	Difference from Zero (*p* Value) †	Mean DA (D @ Degree)	Mean DA Magnitude (D) ± SD	Range of DA (D)	Percentage of DA within 0.50 D
Automated KA		0.26 ± 0.37	<0.001	−0.03 ± 0.32	0.784	0.30 @ 177	0.49 ± 0.24	0.00–1.07	50.0
Scheimpflug KA		0.09 ± 0.35	0.209	−0.06 ± 0.44	0.477	0.37 @ 178	0.53 ± 0.19	0.10–0.91	47.1
Scheimpflug TCRP 2.0 mm	pupil/zone	−0.19 ± 0.57	0.066	−0.01 ± 0.53	0.859	0.17 @ 170	0.68 ± 0.42	0.12–2.03	41.2
	apex/zone	−0.46 ± 0.58	<0.001	−0.01 ± 0.45	0.822	0.17 @ 89	0.72 ± 0.47	0.12–2.03	38.2
	pupil/ring	−0.22 ± 0.42	0.010	0.04 ± 0.38	0.445	0.10 @ 53	0.52 ± 0.31	0.05–1.50	52.9
	apex/ring	−0.23 ± 0.41	0.004	0.04 ± 0.44	0.449	0.07 @ 38	0.55 ± 0.33	0.06–1.50	52.9
Scheimpflug TCRP 3.0 mm	pupil/zone	−0.16 ± 0.38	0.043	0.02 ± 0.38	0.487	0.07 @ 47	0.47 ± 0.29	0.05–1.15	55.9
	apex/zone	−0.14 ± 0.40	0.074	0.07 ± 0.45	0.268	0.12 @ 18	0.54 ± 0.29	0.06–1.15	58.8
	pupil/ring	−0.13 ± 0.38	0.089	0.16 ± 0.43	0.048	0.29 @ 13	0.53 ± 0.29	0.03–1.38	52.9
	apex/ring	−0.19 ± 0.41	0.008	0.14 ± 0.38	0.042	0.18 @ 29	0.51 ± 0.31	0–1.39	52.9
Scheimpflug TCRP 4.0 mm	pupil/zone	−0.05 ± 0.30	0.478	0.05 ± 0.34	0.334	0.11 @ 22	0.40 ± 0.20	0.03–0.85	67.6
	apex/zone	−0.04 ± 0.33	0.407	0.09 ± 0.40	0.184	0.20 @ 15	0.47 ± 0.23	0.02–1.07	50.0
	pupil/ring	−0.18 ± 0.55	0.077	0.09 ± 0.48	0.221	0.39 @ 11	0.67 ± 0.32	0.17–1.36	38.2
	apex/ring	−0.17 ± 0.48	0.032	0.08 ± 0.45	0.327	0.26 @ 9	0.59 ± 0.32	0.11–1.20	44.1
Difference between parameters (*p* value) *		<0.001					<0.001		

SD: standard deviation; KA: keratometric astigmatism; TCRP: total corneal refractive power; * *p* value by the Friedman test comparing average values of all corneal measurements; † *p* value by one-sample Wilcoxon signed rank test.

**Table 7 diagnostics-13-03687-t007:** Linear regression equations for refractive astigmatism in 95 eyes. These correlations are based on Equation (7).

Keratometric Measurement		Linear Regression Equation	*r* ^2^	* *p* Value
Automated KA		−0.04 + 0.92 KP(Φ) + 0.39 cos(2α)	0.76	<0.001
Scheimpflug KA		−0.17 + 0.99 KP(Φ) + 0.36 cos(2α)	0.81	<0.001
Scheimpflug TCRP 2.0 mm	pupil/zone	−0.13 + 0.78 KP(Φ) + 0.09 cos(2α)	0.50	<0.001
	apex/zone	−0.23 + 0.71 KP(Φ) + 0.04 cos(2α)	0.50	<0.001
	pupil/ring	−0.22 + 0.86 KP(Φ) + 0.07 cos(2α)	0.66	<0.001
	apex/ring	−0.26 + 0.90 KP(Φ) + 0.05 cos(2α)	0.66	<0.001
Scheimpflug TCRP 3.0 mm	pupil/zone	−0.24 + 0.91 KP(Φ) + 0.07 cos(2α)	0.69	<0.001
	apex/zone	−0.22 + 0.91 KP(Φ) + 0.09 cos(2α)	0.68	<0.001
	pupil/ring	−0.24 + 0.95 KP(Φ) + 0.14 cos(2α)	0.67	<0.001
	apex/ring	−0.15 + 0.84 KP(Φ) + 0.12 cos(2α)	0.68	<0.001
Scheimpflug TCRP 4.0 mm	pupil/zone	−0.13 + 0.93 KP(Φ) + 0.13 cos(2α)	0.77	<0.001
	apex/zone	−0.16 + 0.94 KP(Φ) + 0.15 cos(2α)	0.71	<0.001
	pupil/ring	−0.29 + 0.94 KP(Φ) + 0.27 cos(2α)	0.63	<0.001
	apex/ring	−0.19 + 0.86 KP(Φ) + 0.26 cos(2α)	0.65	<0.001

KA: keratometric astigmatism; EKR: equivalent K-reading; TCRP: total corneal refractive power; * *p* value by multiple linear regression test.

**Table 8 diagnostics-13-03687-t008:** Difference in astigmatism (DA) derived from the refractive cylinder and *optimized value* of corneal astigmatism in the whole sample (*n* = 95). Calculations are based on polar values along *variable* meridians in the form of the steep anterior corneal meridian for each eye and for each measuring modality.

Keratometric Measurement		∆KP(Φ) (D) ± SD	Difference from Zero (*p* Value) †	∆KP(Φ+45) (D) ± SD	Difference from Zero (*p* Value) †	Mean DA (D @ Degree)	Mean DA Magnitude (D) ± SD	Range of DA (D)	Percentage of DA Magnitude within 0.50 D
Automated KA		0.00 ± 0.32	0.509	−0.04 ± 0.31	0.579	0.04 @ 133	0.38 ± 0.23	0.03–1.30	71.6
Scheimpflug KA		0.00 ± 0.30	0.844	−0.01 ± 0.37	0.799	0.01 @ 146	0.42 ± 0.21	0.02–0.94	67.4
Scheimpflug TCRP 2.0 mm	pupil/zone	0.00 ± 0.47	0.994	0.04 ± 0.42	0.189	0.04 @ 43	0.53 ± 0.33	0.01–1.60	54.7
	apex/zone	0.00 ± 0.48	0.994	0.06 ± 0.37	0.077	0.06 @ 44	0.53 ± 0.29	0.01–1.39	50.5
	pupil/ring	0.00 ± 0.39	0.724	0.08 ± 0.32	0.017	0.08 @ 46	0.45 ± 0.23	0.02–1.14	65.3
	apex/ring	0.00 ± 0.39	0.897	0.10 ± 0.37	0.007	0.10 @ 45	0.47 ± 0.26	0.04–1.31	64.2
Scheimpflug TCRP 3.0 mm	pupil/zone	0.00 ± 0.37	0.622	0.07 ± 0.32	0.018	0.07 @ 42	0.44 ± 0.22	0.03–1.02	65.3
	apex/zone	0.00 ± 0.38	0.867	0.11 ± 0.36	0.004	0.11 @ 44	0.47 ± 0.25	0.04–1.19	57.9
	pupil/ring	0.00 ± 0.38	0.667	0.13 ± 0.34	0.001	0.13 @ 44	0.46 ± 0.25	0–1.10	58.9
	apex/ring	0.00 ± 0.37	0.758	0.11 ± 0.32	0.002	0.11 @ 46	0.44 ± 0.25	0.01–1.09	63.2
Scheimpflug TCRP 4.0 mm	pupil/zone	0.00 ± 0.31	0.683	0.07 ± 0.30	0.013	0.07 @ 45	0.38 ± 0.21	0.02–0.88	73.7
	apex/zone	0.00 ± 0.35	0.705	0.10 ± 0.33	0.007	0.10 @ 46	0.43 ± 0.23	0.03–0.99	61.1
	pupil/ring	0.00 ± 0.42	0.956	0.08 ± 0.38	0.059	0.08 @ 45	0.50 ± 0.28	0.01–1.14	55.8
	apex/ring	0.00 ± 0.40	0.962	0.05 ± 0.39	0.338	0.05 @ 46	0.49 ± 0.27	0.01–1.27	54.7
Difference between parameters (*p* value) *		0.746		0.027			<0.001		

SD: standard deviation; KA: keratometric astigmatism; EKR: equivalent K-reading; TCRP: total corneal refractive power; * *p* value by the Friedman test comparing average values of all corneal measurements; † *p* value by one-sample Wilcoxon signed rank test.

**Table 9 diagnostics-13-03687-t009:** Comparison of difference in astigmatism (DA) according to optimization. DA was provided by the automated keratometer and Scheimpflug TCRP 4.0 mm (pupil/zone) in the whole sample (*n* = 95).

Keratometric Measurement		∆KP(Φ) (D) ± SD	Difference between Automated KA and TCRP 4.0 mm, Pupil/Zone (*p* Value) *	∆KP(Φ+45) (D) ± SD	Difference between Automated KA and TCRP 4.0 mm, Pupil/Zone (*p* Value) *	Mean DA Magnitude (D) ± SD	Difference between Parameters (*p* Value) *
Before optimization	Automated KA	−0.13 ± 0.46	0.662	−0.04 ± 0.31	0.003	0.51 ± 0.25	0.001
	TCRP 4.0 mm Pupil/zone	−0.16 ± 0.33		0.07 ± 0.30		0.42 ± 0.24	
After optimization	Automated KA	0.00 ± 0.32	0.499	−0.04 ± 0.31	0.003	0.38 ± 0.23	0.733
	TCRP 4.0 mm Pupil/zone	0.00 ± 0.31		0.07 ± 0.30		0.38 ± 0.21	
Difference according to optimization (*p* value) *	Automated KA	<0.001		1.00		<0.001	
	TCRP 4.0 mm Pupil/zone	<0.001				0.054	

SD: standard deviation; KA: keratometric astigmatism; EKR: equivalent K-reading; TCRP: total corneal refractive power; * *p* value by the Wilcoxon signed ranked test.

## Data Availability

The data presented in this study are available on request from the corresponding author. The data are not publicly available due to patients’ privacy and policies of hospitals that approve IRB.

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
