# Peer review of "Comparison of Automated Keratometer and Scheimpflug Tomography for Predicting Refractive Astigmatism in Pseudophakic Eyes"

_diagnostics, 2023, doi:10.3390/diagnostics13243687_

Round 1

Reviewer 1 Report

Comments and Suggestions for Authors

This study analyzes the correspondence between refractive astigmatism and corneal astigmatism in pseudophakic eyes with non-toric intraocular lenses by comparison of automated keratometer and Scheimpflug tomography. It is well written and designed.

The major concern is why not do the same comparison in patient with phakia eye? Because the goal is to determine which value is most closely associated with the refractive cylinder and to calculate the power of toric intraocular lenses (IOLs) before cataract surgery.

Author Response

Thank you for bringing up a valid point. In this study, we did not include an analysis of crystalline lenses, but only analyzed pseudophakic eyes with little tilting or decentration.
If we had analyzed only clear lenses without cataract progression, it would have been appropriate to analyze phakic eyes. However, since lenticular astigmatism occurs in eyes with actual cataract progression, and the effect of lenticular astigmatism disappears after cataract extraction, we did not investigate phakic eyes in this study.

Reviewer 2 Report

Comments and Suggestions for Authors

It's a very good work, well planned and realized, with good statistical analysis and interesting results. I recommend it for pubblication without any revision.

Author Response

Thank you for kind comment.

Reviewer 3 Report

Comments and Suggestions for Authors

The subject of the study is actually of very low interest and significance since it is a simple comparison of clinical Vs. machinery measurements. 

The results are almost obvious.

Nevertheless, it is very well written.  

Author Response

Thank you for your kind comments.

Reviewer 4 Report

Comments and Suggestions for Authors

The manuscript entitled Comparison of automated keratometer and Scheimpflug tomography for predicting refractive astigmatism in pseudophakic eyes, brings a new inside in the diagnosis of corneal astigmatism and calculation of toric intraocular lens in pseudophakic eyes.

Observations

Introduction

Please indicate in details what is keratometric index, how is calculated and how is related with total corneal power. 

Please offer more details about the utility and accurate measurements of corneal astigmatism.

Indicate the side effects of miscalculation of the power of toric intraocular lens.

Indicate what is TCA before abbreviation.

Material and Methods.

Explain why you choose the visual acuity higher than 20/40 as reference in this study. Was a random value or you had some reasons to choose this value?

Discussions

Please organize your discussions according with other studies. There are only few references  of other similar studies, where you discuss your results compared with other research in the area.

Conclusions

Indicate the utility of your study.

Round 2

Reviewer 1 Report

Comments and Suggestions for Authors

no more comment